# Eye-head coordination and dynamic visual scanning as indicators of visuo-cognitive demands in driving simulator

**Laura Mikula** [1]*, **Sergio Mejía-Romero** [1], **Romain Chaumillon** [1], **Amigale Patoine** [1], **Eduardo Lugo** [1], **Delphine Bernardin** [1,2], **Jocelyn Faubert** [1]

**1** Faubert Laboratory, School of Optometry, Université de Montréal, Montréal, Québec, Canada, **2** Essilor International, Research and Development Department, Paris, France & Essilor Canada, Saint-Laurent, Canada

* laura.mikula@umontreal.ca

**Data Availability Statement:** Relevant files are available from the Open Science Framework repository (https://osf.io/da9yp/).

**Funding:** This work was supported by the Natural Sciences and Engineering Research Council of

## Abstract

Driving is an everyday task involving a complex interaction between visual and cognitive processes. As such, an increase in the cognitive and/or visual demands can lead to a mental overload which can be detrimental for driving safety. Compiling evidence suggest that eye and head movements are relevant indicators of visuo-cognitive demands and attention allocation. This study aims to investigate the effects of visual degradation on eye-head coordination as well as visual scanning behavior during a highly demanding task in a driving simulator. A total of 21 emmetropic participants (21 to 34 years old) performed dual-task driving in which they were asked to maintain a constant speed on a highway while completing a visual search and detection task on a navigation device. Participants did the experiment with optimal vision and with contact lenses that introduced a visual perturbation (myopic defocus). The results indicate modifications of eye-head coordination and the dynamics of visual scanning in response to the visual perturbation induced. More specifically, the head was more involved in horizontal gaze shifts when the visual needs were not met. Furthermore, the evaluation of visual scanning dynamics, based on time-based entropy which measures the complexity and randomness of scanpaths, revealed that eye and gaze movements became less explorative and more stereotyped when vision was not optimal. These results provide evidence for a reorganization of both eye and head movements in response to increasing visual-cognitive demands during a driving task. Altogether, these findings suggest that eye and head movements can provide relevant information about visuo-cognitive demands associated with complex tasks. Ultimately, eye-head coordination and visual scanning dynamics may be good candidates to estimate drivers' workload and better characterize risky driving behavior.

## Introduction

Vision is one of the most important sensory input used when driving [1], and as such intact visual processing and functions are a prerequisite for driver's license application. The most

Canada, NSERC – Essilor Industrial Research Chair (IRCPJ 305729-13), Research and development cooperatif NSERC – Essilor Grant (CRDPJ 533187 - 2018), Prompt (https://www.nserc-crsng.gc.ca/index_eng.asp & https://www.essilor.ca). The funders provided support in the form of salary for author DB but had no role in study design, data collection and analysis, decision to publish, or preparation of the manuscript. Authors LM and AP received support from student awards from the Road Safety Research Network (Réseau de Recherche en Sécurité Routière) of Québec (https://rrsr.ca/en).

**Competing interests:** I have read the journal's policy and the authors of this manuscript have the following competing interests: author DB is an adjunct professor of the NSERC/Essilor Chair, RDC NSERC-Essilor project at Université de Montréal and is employed by Essilor Canada as research project manager. This does not alter our adherence to PLOS ONE policies on sharing data and materials.

widespread vision assessment comprises measurements of visual acuity and the extent of visual field. However, not all drivers are subject to the same regulations since legal visual requirements for driving can vary between, and sometimes within, countries. For instance, the minimum visual acuity imposed in the United States ranges between 20/40 and 20/100 depending on the jurisdiction [2], whereas Canadian vision standards are set to 20/50 except for the provinces of New Brunswick and Nova Scotia which require a minimum of 20/40, in accordance with the recommendation of the International Council of Ophthalmology [3,4].

Although focus has been primarily directed to the vision standards for driving, to date studies have shown relatively weak or inconsistent relationships between stationary visual acuity by itself and motor vehicle collision involvement [1,5–10]. Besides, the likelihood of road accidents is not increased in drivers with visual acuity less than 20/40 compared to those with better vision [8,11]. Given the highly visual complexity of driving, it is not surprising that visual acuity alone does not reflect all the capacities necessary to safely operate a motor vehicle.

In contrast, visual attention and cognitive functions appear to be better predictors of driver safety behavior. The Useful Field of View (UFOV) test for example, has been designed to evaluate visual speed processing and divided as well as selective attention. Previous research has revealed that the impaired performance on the UFOV observed in older adults is indeed related to greater crash risk [5,10,12–14] and poorer driving performance [15–17]. Similarly, it has been reported that diminished cognitive functions are associated with unsafe driving [18–20]. And more recently, perceptual-cognitive abilities measured using 3-Dimensional Multiple Object Tracking (3D-MOT) have been shown to predict the performance of older adults in driving simulator [21,22].

Therefore, driving is a cognitively demanding task that requires not only proper visual but also cognitive functions. As a result of the limited human brain capacity, the increased mental workload associated with multitasking results in impaired behavioral performance [23–25]. Distracted driving refers to any concurrent activity that can withdraw attention from the primary task of driving and includes but is not limited to eating, texting, interacting with passengers or using entertainment and navigation systems [26]. Indeed, about 21% of fatalities and 27% of serious injuries that happened in 2016 involved some form of distracted driving according to Transport Canada's National Collision Database. The use of in-vehicle devices has been shown to increase drivers' cognitive resources, resulting in a reduced ability to control their car [27–29]. Performing concurrent tasks while driving is thus a major concern for road safety as it is likely to disrupt visual attention allocation [30–32].

The coupling between eye movements and attention has been well-documented, showing that attention is deployed to the location of a future saccade [33–35]. As a consequence of greater mental workload, the attentional resources available to visually explore the environment are drastically reduced. In the context of driving, it results in spatial gaze concentration as well as more frequent and longer fixations away from the road, which can affect the detection of potential hazards [36–39]. In addition to classical eye movement metrics, including fixation rate and duration, entropy has been derived from information theory [40] to provide a quantitative analysis of gaze behavior in naturalistic environments such as flight simulation [41–44], surgery [45–47] and driving [48–51]. The entropy captures visual scanning complexity and, by extension, the spatial distribution of visual attention using two measures. The stationary entropy characterizes the overall gaze dispersion across the visual scene whereas the transition entropy reflects the randomness (or the predictability) in the scanning pattern [52–54].

Previous research has shown decreased gaze entropy in pilots during high-complexity flights [42,44] and when performing a subsidiary task while driving [48]. Thus, increasing cognitive demands seem to be associated with lower entropy, which indicates less exploration and

a more predictable pattern of visual scanning [54]. In contrast, more traditional eye metrics such as saccade amplitude and fixation duration were not found to be as sensitive to the modulation of mental workload in these situations. Furthermore, pilots' electroencephalographic (EEG) recordings revealed that the reduction of gaze entropy induced by flight complexity is accompanied by enhanced frontal theta oscillations [41]. Interestingly, EEG frontal theta power is known to be a neurophysiological index of cognitive and attentional demands [55–58]. These results suggest that entropy measures reflect changes of the ocular behavior in response to the task workload. However, it should be noted that the calculation of transition entropy in the aforementioned studies relies on the spatial discretisation of the visual space and is therefore highly dependent on the visual content of the scene. This can be problematic in the case of dynamically changing stimuli that can modify fixation probabilities and thus entropy measures. An alternative method that has been put forward this limitation is to investigate eye movements over multiple temporal segments, instead of spatial areas of interest [59]. The benefit of this time-based entropy measure is that it is less sensitive to shifts of the visual scene in more naturalistic settings, where head and body movements occur.

When driving, as opposed to less naturalistic environments, head movements are likely to be involved [60]. Indeed, gaze shifts greater than 15˚ usually rely on a combination of both eye and head movements [61–64]. Moreover, it has been shown that the processing of visual stimuli is enhanced when presented in the straight-ahead direction, independently of gaze direction [65,66]. These results suggest that there is a relationship between visuospatial attention and head movements, as was described for eye movements. Using a driving simulator, different eye-head coordination dynamics have been recorded following exogenous (i.e., stimulus-driven) and endogenous (i.e., goal-oriented) attention shifts [67]. Furthermore, progressive lens wearers have been shown to modify their eye-head coordination in real-world driving [68] or when presented with driving-related scenes [69], providing evidence that eye-head movements are modulated by the quality of visual inputs. Altogether, these findings show that coordinated movements of the eyes and head are related to visuo-cognitive processing [70].

As stated above, driving is a complex activity and if the associated visuo-cognitive needs are not met, driver's safety can be compromised. The present study aims to investigate the effects of increasing visual and cognitive demands on eye-head coordination as well as visual scanning behavior in a driving simulator scenario. The driving task load was manipulated by introducing a concurrent visual search and detection task while visual demands were challenged by an experimental visual degradation induced by contact lens wear. The spatial distributions of eye-head movements were analyzed as well as the coordination between the eyes and the head. For this study, in contrast to traditional entropy measures which describe the spatial distribution of movements, entropy was computed based on temporal dynamics. Consequently, time-based entropy metrics were implemented for the gaze, but also the eyes and the head signals in order to better describe the changes in scanning behavior as a result of greater visuo-cognitive demands.

## Materials and methods

### Participants

Twenty-one participants (15 males, 6 females) aged between 21 and 34 years old (mean ± SD = 24.8 ± 3.7) were recruited for this study. They all had a valid driver's license for at least 5 years and they had normal vision or corrected with contact lenses (i.e., far visual acuity of 4/10 or better in ETDRS chart, stereoscopic acuity of minimum 50 seconds of arc in Randot test and a visual field of at least 100 continuous degrees along the horizontal meridian and at least 10 continuous degrees above and 20 continuous degrees below fixation). All participants had myopia and hyperopia lower than 3.00 diopters (D), maximum astigmatism of 0.75

D and anisometropia lower than 1.00 D. Participants did not suffer from any visual, vestibular, neurological, musculoskeletal or cardiovascular impairment. In addition, scores on the UFOV test showed that all individuals were classified in the low or very-low risk groups for probability of driving difficulties. All participants provided informed written consent and received a compensation of 15$ after the completion of the experiment. This study and all the experimental procedures were approved by the ethics committee of Université de Montréal (Comité d'éthique de la recherche clinique CERC; certificate N˚18-090-CERES-D).

## Apparatus

The Virage VS500M car driving simulator (Virage Simulation Inc.®, Montréal, Canada) was used for the driving tasks. The cockpit consisted of real car parts including a seat, a force feedback steering wheel, accelerator and brake pedals, a dashboard and automatic transmission and controls. The visualization system included a PC 5-channel image generator as well as three 50-inch high-resolution (1280 x 720 pixels) overhead projection screens providing 180˚ front views. In addition, two smaller lateral screens were positioned in the back to replicate blind spot and mirror visualization. Rearview and side mirrors were projected on the central screen at their approximate locations in a real car. The driving cabin of the simulator was mounted on a three-axis platform with electric actuators that recreated the haptic feedback of acceleration, braking, engine-induced vibrations and road texture. A stereo sound system provided realistic engine and external road sounds, and Doppler effects were used to generate naturalistic sounds of surrounding traffic. All auditory information was adjusted to the driving speed. The navigation device consisted of an 8-inch LCD monitor (1024 x 768 pixels) which was located on the car center console, at the level of the air vents and approximately 70 cm away from participants' head.

Head movements were recorded using an OptiTrack motion capture system and the Motive software (NaturalPoint Inc., Oregon, USA) at a sampling rate of 120 Hz. Participants wore a helmet with 4 retro-reflective markers located above the head. The 3D positions of the markers were recorded by a camera placed on top of the simulator's center screen, in order to track yaw, pitch and roll rotations of the head while participants were driving. Eye movements were recorded using SMI Eye Tracking Glasses 2 Wireless (SensoMotoric Instruments, Teltow, Germany) which is a mobile eye-tracking system providing native, binocular tracking with a sampling rate of 120 Hz. The infrared light emitted by the glasses allowed two cameras located in the lower frame to track the corneal reflections and determine the center of the pupils. Eye rotations were recorded in yaw and pitch after a standard 3-point calibration.

Both eye and head recordings were synchronized in time. In order to estimate the gaze-in-space rotation, eye-in-head orientation recorded by the eye-tracker was combined with head-in-space position measured by the motion capture system. These coordinate system transformations required additional calibration procedures. First, the position of the eye rotation center relative to the head and the wearable eye-tracker was computed based on pictures taken at different angles [71]. Then, participants were instructed to move their eyes, but not their head, to fixate a set of 18 points displayed on the central screen of the driving simulator. This particular procedure ensures an accurate estimation of gaze position in 3D world coordinates and allows the computation of intersections with the driving simulator screen and the navigation device on the center console.

## Procedure

In the driving scenario, participants were on a highway and the task consisted of maintaining their speed at 90 km/h, as accurately as possible. They were instructed to drive normally, as

they would in real life and respect road signs, speed limits and other road users. While driving, participants were asked to perform a visual search task on a navigation device located in periphery, on the center console. One trial of the visual search task proceeded as follows. The navigation device turned on and displayed a picture of direction road signs with multiple information (Fig 1A) for 6 seconds and then the device was turned off. Participants had to

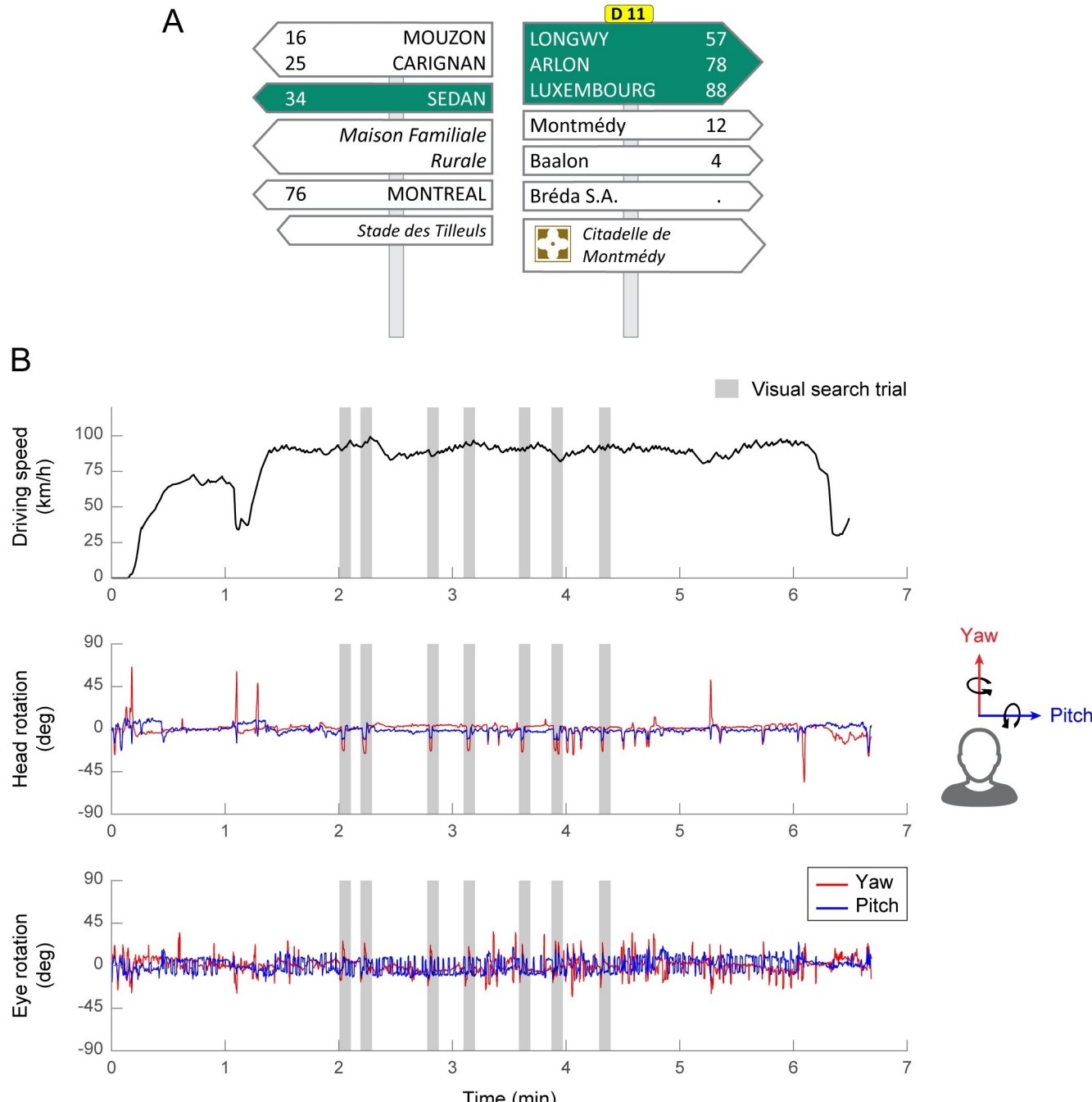

**Fig 1. A. Example of a direction road sign presented on the navigation device during the visual search task.** Participants were asked to give the number associated with the city "Montréal" (here 76). **B. Temporal sequence of the driving scenario.** Driving speed, head rotation and eye rotation as a function of time. Grey shaded areas depict the 7 visual search trials. Yaw rotations are represented in red and pitch rotations in blue.

verbally report the number associated with the city "Montréal" which appeared on each road signs among other directions. They were not given any cue to the onset of a given trial; however, no time limit was imposed and they were allowed to respond during and after the stimuli presentation. A total of 7 visual search trials were randomly distributed in time throughout the driving scenario, which lasted approximately 6 minutes and 30 seconds (Fig 1B). In total, 14 different pictures could be presented to participants so that they did not see the same road sign twice.

Each participant performed the experiment in two conditions: with normal or corrected-to-normal vision (Optimal vision) and with reduced visual acuity (Degraded vision). The order of both experimental conditions was counterbalanced across participants to control for practice effects. For the degraded vision condition, participants were divided into two different groups. In the "lower degradation" group, the contact lenses power was chosen so that participants' visual acuity would approach 4/10 at 3 m distance whereas for the "higher degradation" group, the targeted visual acuity was still 4/10 but at a distance of 1m20. Eleven out of the 21 participants (8 males, 3 females; age = 24.2 ± 3.5 years old) were in the lower perturbation group and the remaining 10 participants (7 males, 3 females; age = 25.5 ± 3.9 years old) were in the higher perturbation group. Disposable soft contact lenses (CooperVision Inc.®) were used to reduce participants' visual acuity. The power of contact lenses was defined by calculating the difference between the target threshold visual acuity and the visual acuity without correction of the participant.

## Variables

Head rotations were recorded in the world coordinate system whereas eye rotations were measured in the head coordinate frame. In addition, gaze (eye-in-space) rotations were computed afterwards by using the head-in-space and the eye-in-head recorded signals. For the purpose of this study, yaw and pitch rotations were analyzed. Negative yaw angles indicate left and positive yaw angles indicate right. On the other hand, negative pitch angles correspond to downward rotations whereas positive pitch rotations describe upward rotations in the corresponding coordinate system.

Eye-head coordination was investigated by considering the linear regression of head versus eye rotations. If the eyes move without any head rotation, the slope would equal 0. In contrast, the bigger the slope, the more involved the head is in coordinated eye-head movements. In both visual conditions, linear regressions slopes were computed for each individual and then averaged across participants within the same degradation group. In order to exclude potential outliers and data artefacts from the regression analysis, data with low-density distribution (less than 20 occurrences per 1x1 degree bin) were removed. Eye-head coordination was analyzed only during the visual search trials, separately for yaw and pitch rotations.

In contrast to eye-head coordination, entropy was evaluated throughout the whole driving scenario, and not only during the visual search task. The X and Y positions of eye, head and gaze were first combined to compute the Euclidean distance of each effector's movement ($\sqrt{X^2 + Y^2}$). In order to evaluate time-based entropy, the eye, head and gaze data were divided into time bins of 120 ms, which approximates the minimum fixation duration. Time-based transition entropy was then calculated by using the conditional entropy equation [72] as follows:

$$Entropy = -\sum_{i=1}^{n} p_i \sum_{j=1}^{n} p(i,j) log_2 p(i,j)$$

Where $p_i$ is the probability of transitioning between the simulator central screen and the navigation device in time bin $i$ and $p(i,j)$ the probability to find the same transition pattern in time bins $i$

and *j*. By analogy to traditional transition entropy, where $p_i$ represents the probability that one fixation is in the area of interest *i* and $p(i,j)$ the probability to transition from area *i* to *j*. The entropy calculated was then normalized by the maximum entropy so that it ranged between 0 and 1. As supported by previous research, higher entropy depicts a more complex and less predictable pattern of visual scanning. On the other hand, lower entropy suggests a more structured and less random pattern of scanning behavior [52–54]. All data processing and computations were performed using a custom-written Matlab® toolbox (The MathWorks, Natick, MA, USA).

## Statistical analyses

In order to compare the distributions of eye, head and gaze rotations between optimal and degraded vision, two-sample Kolmogorov-Smirnov tests were used, where the null hypothesis assumes that the two samples come from the same continuous distribution. Two-way mixed ANOVAs were conducted to explore the effect of the visual condition (within-subjects factor–optimal and degraded vision) and the severity of the visual perturbation (between-subjects factor–lower and higher degradation groups). Non-normal data were corrected for distribution skew using the Box-Cox transformation [73]. Tukey's HSD post-hoc tests were used to further explore significant main effects and interactions. Statistical threshold was set to 0.05.

## Results

First of all, we made sure that participants were completing the visual search task on the navigation device. This was confirmed by the overall success rate which was 84.4 ± 3.4% (mean ± SEM; optimal vision, lower degradation group: 92.2 ± 3.0%; degraded vision, lower degradation group: 71.4 ± 7.5%; optimal vision, higher degradation group: 85.7 ± 8.5%; degraded vision, higher degradation group: 88.6 ± 5.9%).

The density distributions of head, eye and gaze rotations during the visual search task are depicted in Fig 2. In optimal vision (Fig 2A), most of the eye, gaze and head rotations in yaw and pitch were centered around the origin (yellow areas). These observations indicate that both the head and the gaze were primarily oriented towards the middle of the central driving simulator screen (i.e., the road ahead). In addition, it shows that the eyes were mostly in the primary position–in other words, the visual axis is parallel to the sagittal plane of the head. Furthermore, the distributions of eye, head and gaze rotations appear to stretch out to the bottom right quadrant (blue green areas) which roughly corresponds to the position of the car center console. This demonstrates that participants were actually involved in the visual search task displayed on the navigation device located in periphery. Interestingly, the distributions of rotations seemed to be modified by the visual degradation induced by the contact lenses (Fig 2B). These differences are better captured through the kernel density estimates depicted in Fig 3. Two-sample Kolmogorov-Smirnov tests revealed that the distributions of rotations between the optimal and degraded vision conditions were significantly different for the eye and the gaze in pitch, and for the head in yaw (all $p < 0.01$). Similar to the optimal vision condition; head, eye and gaze rotations in degraded vision were mostly aligned with the origin. This shows that, irrespective of the visual perturbation, participants tend to keep their head and eyes directed towards the road straight ahead while driving and performing the concurrent visual search task in periphery. However, as a result of the visual degradation, rotations around 0 were reduced and more rotations were observed around -12˚ in pitch for the eyes (Fig 3A), +20˚ in yaw for the head (Fig 3B) and -15˚ in pitch for the gaze (Fig 3C). These findings suggest that, in presence of the visual degradation, participants are more likely to direct their eyes and head towards the navigation device located on the car center console.

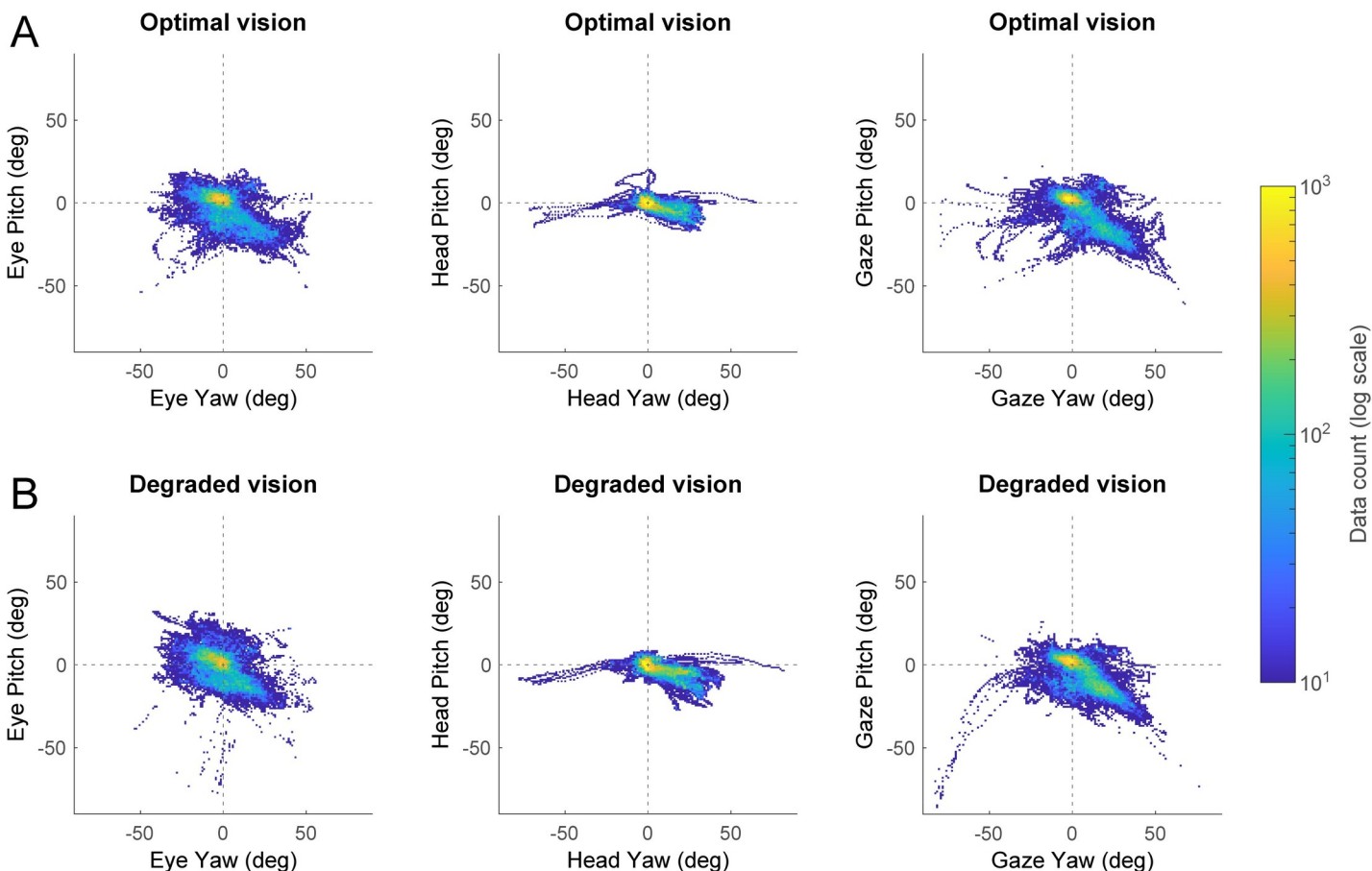

**Fig 2. Density distributions of rotations.** In optimal vision (A) and degraded vision (B) during the visual search task. Pitch rotations are plotted as a function of yaw rotations for the eye (left column), the head (middle column) and the gaze (right column). The color scale depicts the number of data points from all participants ($n$ = 21) for a given combination of yaw-pitch angles.

The coordination between eye and head rotations was first analyzed in the yaw axis (Fig 4). In the optimal vision condition, the average slope of the linear regression between head yaw and eye yaw was -0.47 ± 0.43 for the lower degradation group and -0.22 ± 0.29 for the higher degradation group. In the degraded vision condition, the slopes were -0.50 ± 0.32 and -0.66 ± 0.43 for the least and most impaired group, respectively. A two-way ANOVA revealed a significant main effect of the visual perturbation showing that the average linear regression slope was steeper in the degraded vision condition compared to the optimal vision condition ($F(1,38) = 5.07$, $p = 0.030$, $\eta^2 = 0.11$). However, there was no significant main effect of the degradation group ($F(1,38) = 0.25$, $p = 0.617$) and no significant interaction ($F(1,38) = 2.98$, $p = 0.092$). The eye-head coordination in the pitch axis is illustrated in Fig 5. When vision was optimal, the average slopes were very close to 0 for both degradation groups (lower degradation group: -0.01 ± 0.09 and higher degradation group: 0.00 ± 0.08). In the degraded vision condition, the slope for the lower degradation group was -0.11 ± 0.21 and -0.07 ± 0.24 for the higher degradation group. The two-way ANOVA conducted showed neither significant main effects of the visual degradation ($F(1,38) = 2.73$, $p = 0.107$) and degradation severity ($F(1,38) = 1.00$, $p = 0.323$), nor a significant interaction ($F(1,38) = 0.28$, $p = 0.598$). These findings suggest that the eye-head coordination in yaw, but not in pitch, is modified in response to the visual degradation introduced. More specifically, when vision is not optimal, more head movements

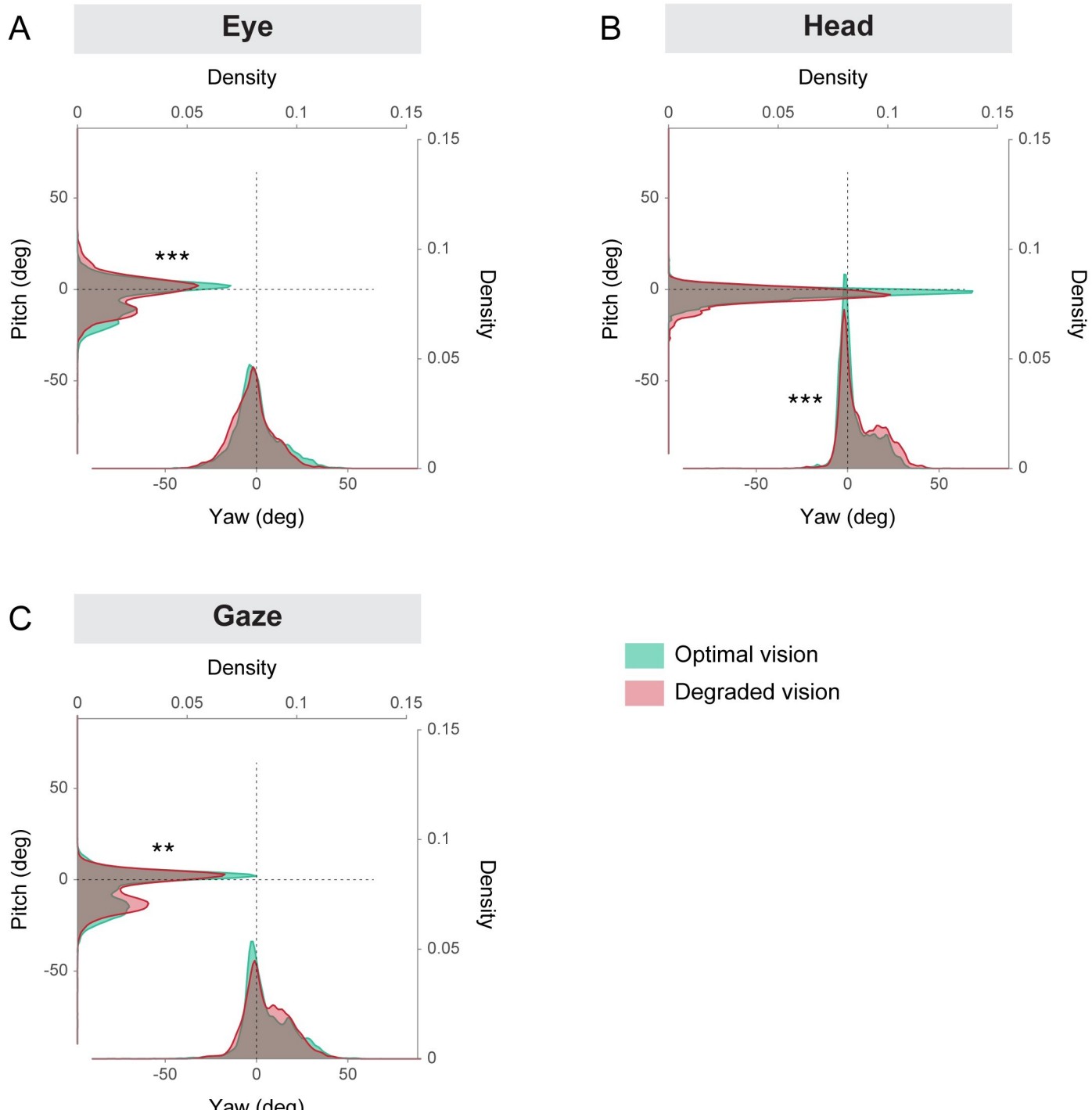

**Fig 3.** Kernel density estimates of eye (A), head (B) and gaze (C) yaw and pitch rotations depicted in Fig 2, in the optimal and degraded vision conditions. $^{**}p < 0.01$, $^{***}p < 0.001$.

are recruited to perform the visual search task while driving. However, eye-head coordination does not seem to reveal differences between the two degradation groups.

In order to further explore the scanning behavior of participants, the time-based entropy was computed across the entire driving scenario and the resulting measures are presented in

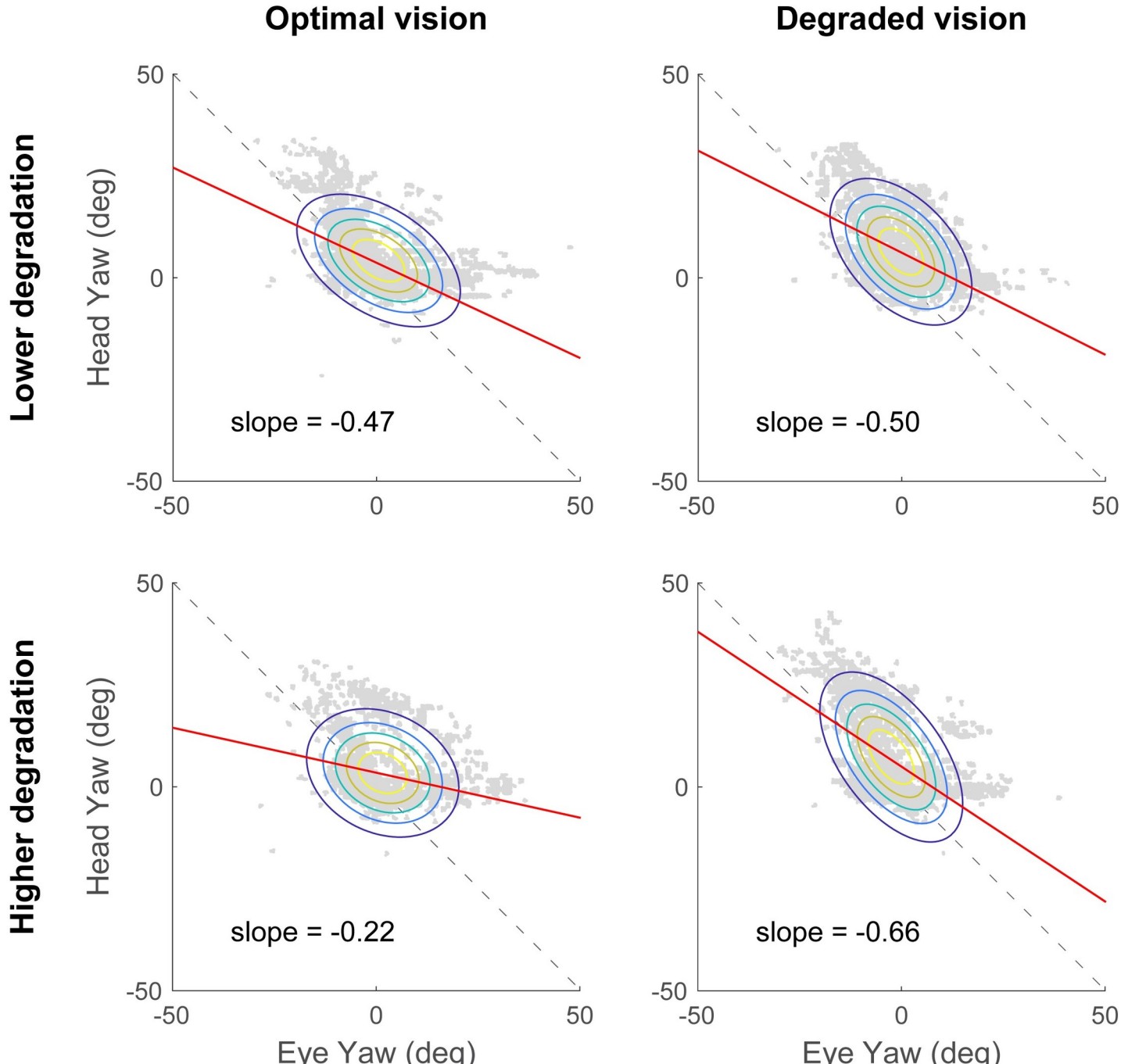

**Fig 4. Eye-head coordination in the yaw axis.** In optimal (right column) and degraded vision (right column) for the lower degradation group (top row) and the higher degradation group (top row) during the visual search task. Head rotations are represented as a function of eye rotations (grey data points). Average slopes of the linear regression (red lines) are reported in each plot. Ellipses represent the joint probability distributions and colored lines correspond to iso-probability contours.

Table 1. Transition entropies for the eye, the head and the gaze are depicted in Fig 6. For the eye signal, the two-way ANOVA revealed a main effect of the visual condition ($F(1,38) = 14.94$, $p < 0.001$, $\eta^2 = 0.25$) showing that eye entropy was significantly reduced in the presence of the visual perturbation, compared to the optimal vision condition. In addition, there was a

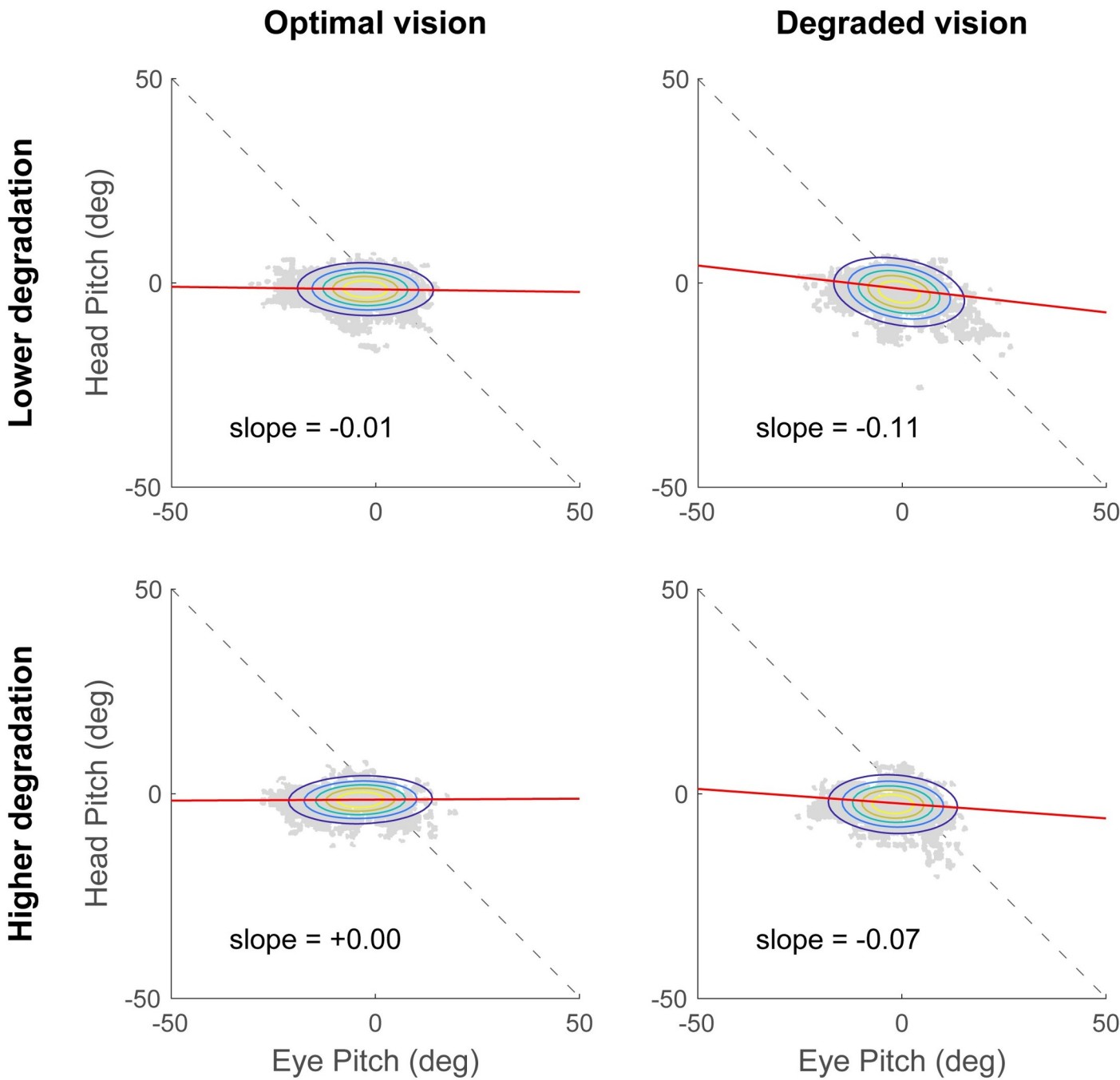

**Fig 5. Eye-head coordination in the pitch axis.** In optimal (right column) and degraded vision (right column) for the lower degradation group (top row) and the higher degradation group (top row) during the visual search task. Head rotations are represented as a function of eye rotations (grey data points). Average slopes of the linear regression (red lines) are reported in each plot. Ellipses represent the joint probability distributions and colored lines correspond to iso-probability contours.

main effect of the degradation severity; the higher degradation group showed reduced eye entropy than the lower degradation group ($F(1,38) = 4.53$, $p = 0.040$, $\eta^2 = 0.08$). One could assume that this effect may be driven by one participant in the lower degradation group having greater eye entropy in degraded vision (Fig 6, top left subplot). However, performing the

**Table 1. Eye, head and gaze entropies.**

| | Optimal vision | | Degraded vision | |
|---|---|---|---|---|
| | Lower degradation | Higher degradation | Lower degradation | Higher degradation |
| **Eye entropy** | 0.477 (± 0.016) | 0.470 (± 0.013) | 0.453 (± 0.044) | 0.422 (± 0.030) |
| **Head entropy** | 0.397 (± 0.010) | 0.394 (± 0.010) | 0.392 (± 0.014) | 0.388 (± 0.007) |
| **Gaze entropy** | 0.476 (± 0.014) | 0.478 (± 0.018) | 0.449 (± 0.047) | 0.458 (± 0.027) |

Average (± standard deviation) entropy values calculated for the eye, the head and the gaze in optimal and degraded vision conditions, for both lower and higher degradation groups.

ANOVA without this participant led to similar results (significant main effect of degradation severity: $F(1,38) = 4.35$, $p = 0.044$, $\eta^2 = 0.06$). In contrast, no significant interaction was reported between visual condition and experimental group ($F(1,38) = 1.76$, $p = 0.192$).

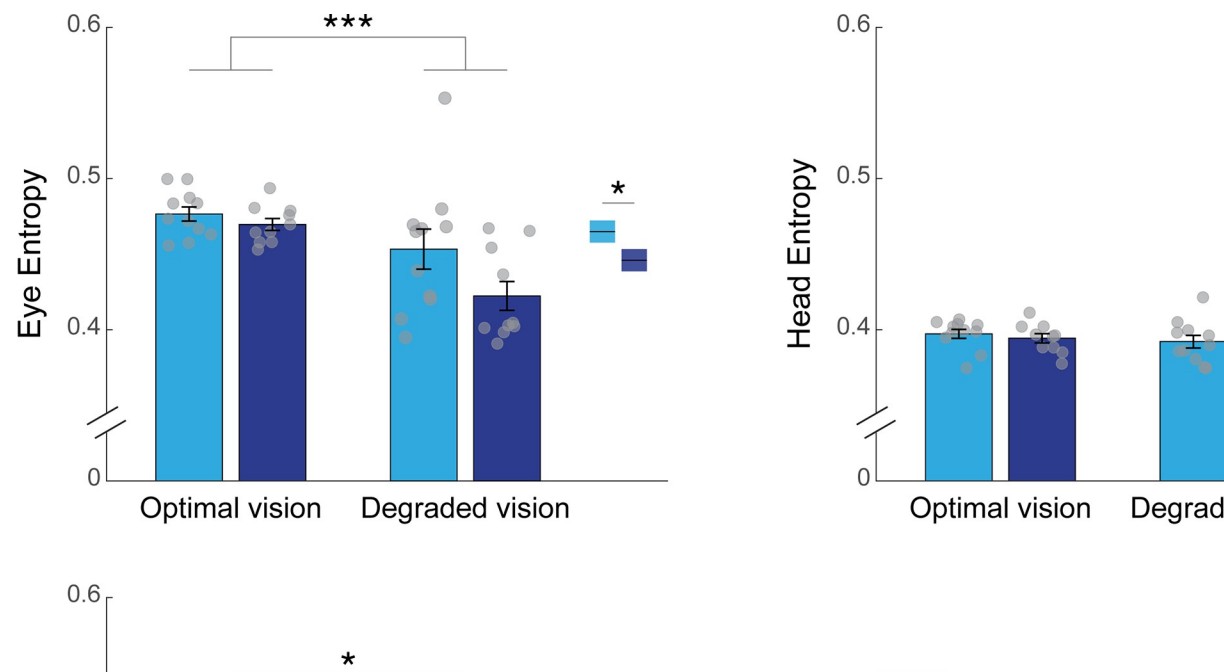

**Fig 6. Entropy values computed for the eye, the head and the gaze as a function of the visual condition and the degradation group.** The histograms represent the average entropy across participants. The error bars represent the SEM and the light grey circles depict the entropy for each individual. Boxes on the right side of each subplot depict the mean entropy for both degradation groups (optimal and degraded vision conditions combined). $^*p < 0.05$, $^{***}p < 0.001$.

Analyses on the head entropy showed no significant main effects of the visual condition ($F$(1,38) = 3.35, $p$ = 0.075) or the severity of the degradation ($F$(1,38) = 1.27, $p$ = 0.267) and no significant interaction ($F$(1,38) = 0.06, $p$ = 0.816). As for the gaze, the ANOVA revealed a significant main effect of the visual condition ($F$(1,38) = 6.69, $p$ = 0.014, $\eta^2$ = 0.15) where entropy was decreased when vision was degraded. However, no main effect of degradation severity ($F$(1,38) = 0.10, $p$ = 0.757) and no significant interaction ($F$(1,38) = 0.08, $p$ = 0.778) were found. These results show that the scanning dynamics of both the eyes and the gaze are similarly affected by the visual perturbation induced.

## Discussion

This study was designed to explore the effects of increasing visual and cognitive demands on eye-head movements and visual scanning pattern while driving. Participants in a driving simulator were asked to maintain constant speed and perform a concurrent visual search and detection task on a navigation device located in the periphery. A visual perturbation was experimentally induced by the wear of contact lenses. The distributions of eye and gaze movements in the pitch axis, and head movements in the yaw axis were found to be affected by the introduction of the visual degradation. Namely, rotations were modified in such a way that participants directed their eyes and head more towards the navigation device on the car center console, where the visual search task was displayed. Nevertheless, in both the optimal and the degraded vision conditions, the head and the gaze were mostly directed on the road ahead as shown by the maxima of density estimates which were centered around 0˚. These observations resemble the findings reporting more gaze concentration towards the road center area as the driving speed or the task difficulty increases [38,74,75]. Furthermore, eye-in-head rotations revealed that the eyes remained somewhat aligned with the front head direction. This is consistent with previous studies that have shown that eye fixations are strongly biased towards the head orientation and that eye-head misalignments can result in impaired visual processing [70,76]. Taken together, these findings suggest that although our participants were performing a visual search in the periphery, their focus remained largely centered on the road in front of the car. One possible explanation could be that participants were able to complete the subsidiary task on the navigation device faster than the maximum 7-second window allowed and, then shifted their eyes and head back to the road for the main driving task. Alternatively, they could have switched multiple times between the road and the navigation device before giving their answer. The examination of reaction times to the visual search task, which we did not measure in the present study, would have been helpful to confirm or refute this hypothesis.

The analysis of eye-head coordination showed that, participants exhibited head movements of greater amplitude in the horizontal plane (i.e., yaw angle), when wearing the contact lenses compared to when vision was optimal. Considering more specifically the degraded vision condition however, no significant difference was observed between the lower and higher degradation groups. Moreover, the eye-head coordination in the pitch axis was not significantly modified when vision was impaired by contact lens wear. These results highlight the specific reorganization of horizontal eye-head movements in response to the visual degradation, while performing a subsidiary visual search task. Our results are in accordance with the observation that participants tend to make more head movements towards the visual stimuli of interest when task difficulty increases [77]. Nevertheless, the slope of the linear regression of head versus eye rotations never exceeded -1. This implies that overall, the eyes contribute more than the head to gaze shifts, as previously described for the discrimination of visual targets in the periphery [78]. Interestingly, the change in eye-head strategy highlighted in the present study echoes earlier work that has reported more and larger vertical head movements in participants

adapting to progressive addition lenses [69,79]. This is attributed to the gradual change in power from the top to the bottom of progressive lenses, necessary to provide clear vision at all distances. In our study, however, horizontal movements were more likely to be involved as the secondary visual search task was displayed on the car center console. That would explain why our visual perturbation specifically affected the eye-head coordination across the horizontal, and not the vertical, plane.

The present study sought to investigate drivers' scanning behavior by means of the entropy. We adopted a new approach which takes into consideration the temporal rather than the standard spatial features of eye and head movements, which are typically used for the calculation of entropy. Our modified entropy measures revealed that the scanning pattern of the head was not strongly affected by the visual perturbation induced by the contact lenses. This difference may be due to slower head movements compared to the eyes, thus requiring the use of longer time windows for the calculation of time-based head entropy. However, for the sake of this study we decided to use the same time window of 120 ms for the eyes, the gaze and the head in order to compare entropy measures between all three effectors. On the other hand, both eye and gaze entropies were found to be significantly decreased when vision was impaired, compared to when it was optimal. The entropy of the eye, but not the gaze, was lower in the most compared to the least impaired group. This result suggests that the eyes' scanning pattern might be more sensitive to various levels of blur and could be a more reliable marker of visuo-cognitive demands. Reduced eye and gaze entropies while wearing contact lenses demonstrate that increasing visual demands altered drivers' visual scanning pattern, which became less explorative and more stereotyped. These results are similar to those reporting a reduction in gaze transition entropy when portions of visual stimuli are blurred [80]. Unfortunately, the authors did not discuss this result and it remains unclear why and how a visual perturbation can influence scanning patterns. Two non-mutually exclusive explanations can be advanced. The first is that transition entropy is dependent on the visual scene complexity, as suggested by Shiferaw and colleagues [54]. In that case, the loss of high spatial frequency content due to the blur induced by the contact lenses would result in reduced visual complexity and thus, in lower transition entropy. The second is that non-optimal vision substantially increases the overall cognitive task load. There again, a reduction in transition entropy is expected as it has been shown to decrease during high-complexity flights, complex pattern recognition tasks or dual-task driving [41,42,44,48,81]. Transition entropy is considered to be an indicator of visual scanning efficiency and to reflect the top-down modulation of gaze control [54]. As a consequence, the reduction in transition entropy related to task difficulty or distraction is likely to demonstrate impaired allocation of resources for gaze control, due to overall greater cognitive demands.

Ultimately, our present findings demonstrate that the calculation of entropy based on temporal characteristics of the scanpath provides some advantages over the more traditional entropy measures considering spatial distribution. Indeed, transition entropy as commonly used to characterize scanning patterns quantifies the transition of fixations between different areas of interest that divide the visual scene [52,53,82]. However, under normal viewing conditions, eye movements are for the most part stimulus-driven and attracted to salient elements of the environment [83–85]. This thus suggests that transition entropy, assessed using regions of interest, might be highly dependent upon the visual scene composition and how the visual space is been discretized [54]. Under those circumstances, it makes it difficult to compare transition entropies between paradigms in which visual stimuli are more dynamic and change over time, such as in driving simulator. Furthermore, it has been acknowledged that fixation durations are also relevant to describe scanning patterns and can vary as a function of task difficulty [86,87]. Hence, the use of time-based entropies would allow to minimize the confounding

effects of various visuospatial task demands [59]. Our modified measures of entropy have been shown to vary in the same way as traditional transition entropy does in aviation and driving [41,42,48], therefore providing evidence that this particular method constitutes a valid approach to quantify visual scanning behavior in complex environments. In addition, transition entropy so far has been derived from gaze shifts which combine eye and head signals and to our knowledge, this is the first study to examine the entropy related to eyes and head, separately. These findings suggest that transition entropy can be applied to eye and head recordings as well and, this would allow a better understanding of the relationship between eye and head scanning behaviors and strategies. Although further research is needed to strengthen the present findings, this is of particular interest for studies using more naturalistic or real-world settings in which the head is not restrained.

In this study, we introduced a blur in emmetropic participants in order to challenge their visual system and increase the visuo-cognitive demands of the task. For this purpose, they wore contact lenses with a positive addition which impose myopic defocus, thus resulting in a reduction of visual acuity. The plus lens caused the image to focus in front of, instead of on, the retina. As a result, a change in accommodative demand occurred leading to a mismatch between the vergence and the accommodative responses, referred to as vergence-accommodation conflict. Furthermore, myopic defocus imposed during development is known to trigger structural changes in eye growth to compensate for the lens-induced refractive errors, as reported in a number of animal species [88–91]. These observations corroborate the effect of defocus on visual demands. In addition, the vergence-accommodation conflict has been shown to interfere with cognitive executive functions [92]. It is speculated that the neural correlates between cognitive control and the vergence-accommodation coupling partially overlap at the level of the frontal and parietal lobes. Consequently, both processes compete for visual attention resources when disrupting the balance between vergence and accommodation. Similarly, visual blur has been related to impaired cognitive functioning [93] and the recognition of blurry objects requires larger field of view as well as longer viewing times [94]. These findings support the need to consider the impact of visual degradation on the interaction between vision and cognition, especially as the simple measure of visual acuity appears to be dependent on the effect of both retinal blur and covert attention [95]. This underscores the importance of the quality of vision during highly demanding cognitive tasks, such as driving for example.

Finally, it has been reported that the visual system shows weaker vergence and accommodation responses to blur compared to disparity cues [96,97]. This suggests that the modulation of the vergence response, such as prism-induced disparity for example, would have a greater impact and would potentially allow to reveal distinct effects as a function of the magnitude of the visual perturbation. Moreover, young adults may have sufficient accommodative reserve to partially compensate for the vergence-accommodation mismatch [96,98,99]. It is thus possible that our participants were able to somewhat compensate, even for the blur induced by the stronger addition. That would explain why, in most cases, we did not observe significant differences between the lower and the higher visual degradation groups, and hence found a very limited effect of the severity of visual degradation. That said, it would be interesting to test older adults in order to investigate whether the decline of the accommodative ability exacerbate the effect of the visual degradation on their eye-head coordination and visual scanning behavior while driving.

## Conclusions

To conclude, this study has provided evidence that when performing a highly demanding driving task, drivers adapt their eye-head coordination in order to meet the increasing visual

demands related to vision degradation. By contrast, the overall spatial distribution of eye and head movements appears to be rather insensitive to perturbations of the visual input. Lastly, time-based transition entropy measures have revealed that the scanning behavior of the eyes and the gaze is modified by the visual perturbation induced. This modification in transition entropy suggests a decline in visual scanning efficiency in response to increased visuo-cognitive demands, which can be potentially detrimental for drivers' safety on the road. These findings demonstrate that quantitative measures of visual scanning provide relevant information about the visuo-cognitive demands and possibly the mental workload involved in performing complex tasks such as driving. If so, time-based entropy could be used in upcoming research to help better identify distraction in fields where traditional entropy measures are currently being used, such as driving, aviation or surgery. Ultimately, dynamic visual scanning assessed through time-based entropy might be able to discriminate between safe and unsafe behaviors during these complex and demanding tasks.

## Author Contributions

**Conceptualization:** Romain Chaumillon, Delphine Bernardin.

**Data curation:** Amigale Patoine.

**Formal analysis:** Laura Mikula, Sergio Mejía-Romero, Romain Chaumillon.

**Funding acquisition:** Delphine Bernardin, Jocelyn Faubert.

**Investigation:** Laura Mikula, Amigale Patoine.

**Methodology:** Laura Mikula, Sergio Mejía-Romero, Romain Chaumillon, Eduardo Lugo.

**Project administration:** Delphine Bernardin.

**Resources:** Delphine Bernardin, Jocelyn Faubert.

**Software:** Sergio Mejía-Romero, Romain Chaumillon.

**Supervision:** Delphine Bernardin, Jocelyn Faubert.

**Validation:** Laura Mikula, Delphine Bernardin, Jocelyn Faubert.

**Visualization:** Laura Mikula, Sergio Mejía-Romero.

**Writing – original draft:** Laura Mikula.

**Writing – review & editing:** Laura Mikula, Sergio Mejía-Romero, Romain Chaumillon, Amigale Patoine, Eduardo Lugo, Delphine Bernardin, Jocelyn Faubert.

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
