## [Decision Letter · Decision Letter 0]

21 Oct 2020

PONE-D-20-29576

Eye-head coordination and dynamic visual scanning as indicators of visuo-cognitive demands in driving simulator

PLOS ONE

Dear Dr. Mikula,

Thank you for submitting your manuscript to PLOS ONE. After careful consideration, we feel that it has merit but does not fully meet PLOS ONE’s publication criteria as it currently stands. Therefore, we invite you to submit a revised version of the manuscript that addresses the points raised during the review process.

We look forward to receiving your revised manuscript.

Kind regards,

Feng Chen

Academic Editor

PLOS ONE

Journal Requirements:

2.Thank you for providing the following Funding Statement: 

[This work was supported by the Natural Sciences and Engineering Research Council of Canada, NSERC – Essilor Industrial Research Chair (IRCPJ 305729-13), Research and development cooperatif NSERC – Essilor Grant (CRDPJ 533187 - 2018), Prompt (https://www.nserc-crsng.gc.ca/index_eng.asp & https://www.essilor.ca). The funders provided support in the form of salary for author DB but had no role in study design, data collection and analysis, decision to publish, or preparation of the manuscript. Authors LM and AP received support from student awards from the Road Safety Research Network (Réseau de Recherche en Sécurité Routière) of Québec (https://rrsr.ca/en).].

We note that one or more of the authors is affiliated with the funding organization, indicating the funder may have had some role in the design, data collection, analysis or preparation of your manuscript for publication; in other words, the funder played an indirect role through the participation of the co-authors.

If the funding organization did not play a role in the study design, data collection and analysis, decision to publish, or preparation of the manuscript and only provided financial support in the form of authors' salaries and/or research materials, please review your statements relating to the author contributions, and ensure you have specifically and accurately indicated the role(s) that these authors had in your study in the Author Contributions section of the online submission form. Please make any necessary amendments directly within this section of the online submission form.  Please also update your Funding Statement to include the following statement: “The funder provided support in the form of salaries for authors [insert relevant initials], but did not have any additional role in the study design, data collection and analysis, decision to publish, or preparation of the manuscript. The specific roles of these authors are articulated in the ‘author contributions’ section.”

If the funding organization did have an additional role, please state and explain that role within your Funding Statement.

Please also provide an updated Competing Interests Statement declaring this commercial affiliation along with any other relevant declarations relating to employment, consultancy, patents, products in development, or marketed products, etc.  

Reviewers' comments:

Reviewer's Responses to Questions

**Comments to the Author**

1. Is the manuscript technically sound, and do the data support the conclusions?

Reviewer #1: Partly

Reviewer #2: Yes

2. Has the statistical analysis been performed appropriately and rigorously? 

Reviewer #1: Yes

Reviewer #2: Yes

3. Have the authors made all data underlying the findings in their manuscript fully available?

Reviewer #1: Yes

Reviewer #2: Yes

4. Is the manuscript presented in an intelligible fashion and written in standard English?

Reviewer #1: Yes

Reviewer #2: Yes

5. Review Comments to the Author

Reviewer #1: Generally, the manuscript is well written. However, revisions are required before acceptance.

1. In the experiment, participants were asked to wear contact lenses with varying degrees of defocusing to achieve visual degradation. The manuscript did not specify the extent to which they affected the increase in drivers' visual demands. No significant difference between the low-degradation and high-degradation groups was observed in the study, assuming that the participants could compensate to some extent. This is not rigorous; is there more detail.

2. 21 participants were selected for the experiment. The degree of astigmatism and other factors also have an impact on the experimental results, such as affecting the anchoring of eye tracker, which was not considered in the experiment.

3. The minimum gaze duration of the eye is 120 ms, therefore, eye and gaze data were divided into time bins of 120 ms. However, whether it is reasonable to divide the head movement data into a time bins of 800 ms? There's no basis here. And the coupling between data is questionable. In the discussion section of experimental results, it is found that drivers focus on the road ahead, assuming that they complete the auxiliary tasks faster than expected, which is lack of experimental verification. Therefore, there should be rigorous consideration in this aspect, which is also helpful for analysis.

4. According to the density distribution of the yaw/pitch Angle data of the driver's head, eyes and gaze, it is not precise enough to infer the location of the driver's gaze, which is two-dimensional. Different drivers test at different locations of the equipment, that is, there is a big difference in the original position in the spatial coordinate system, there is an individual difference in the projected position in front of them.

Reviewer #2: The topic is worthy of investigating and the conclusion is useful. The paper is overall well written and structured. There is one minor requested revision to the current paper, there literature review should be exhaustive and include the following driving simulator related studies:

[1] Examining the safety of trucks under crosswind at bridge-tunnel section: A driving simulator study, Tunnelling and Underground Space Technology, 2019, 92, 103034. https://doi.org/10.1016/j.tust.2019.103034

[2] Examining the influence of decorated sidewaall in road tunnels using fMRI technology, Tunnelling and Underground Space Technology, Volume 99, 2020, https://doi.org/10.1016/j.tust.2020.103362

6. PLOS authors have the option to publish the peer review history of their article (what does this mean?). If published, this will include your full peer review and any attached files.

Reviewer #1: No

Reviewer #2: No

---

## [Author Response · Author response to Decision Letter 0]

27 Nov 2020

Response to reviewers has been provided as a separate document

---

## [Decision Letter · Decision Letter 1]

17 Dec 2020

Eye-head coordination and dynamic visual scanning as indicators of visuo-cognitive demands in driving simulator

PONE-D-20-29576R1

Dear Dr. Mikula,

We’re pleased to inform you that your manuscript has been judged scientifically suitable for publication and will be formally accepted for publication once it meets all outstanding technical requirements.

Kind regards,

Feng Chen

Academic Editor

PLOS ONE

Additional Editor Comments (optional):

Reviewers' comments:

Reviewer's Responses to Questions

**Comments to the Author**

1. If the authors have adequately addressed your comments raised in a previous round of review and you feel that this manuscript is now acceptable for publication, you may indicate that here to bypass the “Comments to the Author” section, enter your conflict of interest statement in the “Confidential to Editor” section, and submit your "Accept" recommendation.

Reviewer #1: All comments have been addressed

Reviewer #2: All comments have been addressed

2. Is the manuscript technically sound, and do the data support the conclusions?

Reviewer #1: Yes

Reviewer #2: Yes

3. Has the statistical analysis been performed appropriately and rigorously? 

Reviewer #1: Yes

Reviewer #2: Yes

4. Have the authors made all data underlying the findings in their manuscript fully available?

Reviewer #1: Yes

Reviewer #2: Yes

5. Is the manuscript presented in an intelligible fashion and written in standard English?

Reviewer #1: Yes

Reviewer #2: Yes

6. Review Comments to the Author

Reviewer #1: 1. The author can further establish a reasonable evaluation system based on existing research, such as how to judge whether the person being tested is distracted through observational data

2. How to discriminate between safe and unsafe behaviors during these complex and demanding tasks.

Reviewer #2: (No Response)

7. PLOS authors have the option to publish the peer review history of their article (what does this mean?). If published, this will include your full peer review and any attached files.

Reviewer #1: No

Reviewer #2: No

---

## [Editor Report · Acceptance letter]

21 Dec 2020

PONE-D-20-29576R1 

Eye-head coordination and dynamic visual scanning as indicators of visuo-cognitive demands in driving simulator 

Dear Dr. Mikula:

I'm pleased to inform you that your manuscript has been deemed suitable for publication in PLOS ONE. Congratulations! Your manuscript is now with our production department. 

Kind regards, 

on behalf of

Dr. Feng Chen 

Academic Editor

PLOS ONE